# Seeing-Eye Quadruped Navigation with Force Responsive Locomotion Control

**David DeFazio**  **Eisuke Hirota**  **Shiqi Zhang**
Binghamton University
{ddefazi1,ehirota1,zhangs}@binghamton.edu

**Abstract:**

Seeing-eye robots are very useful tools for guiding visually impaired people, potentially producing a huge societal impact given the low availability and high cost of real guide dogs. Although a few seeing-eye robot systems have already been demonstrated, none considered external tugs from humans, which frequently occur in a real guide dog setting. In this paper, we simultaneously train a locomotion controller that is robust to external tugging forces via Reinforcement Learning (RL), and an external force estimator via supervised learning. The controller ensures stable walking, and the force estimator enables the robot to respond to the external forces from the human. These forces are used to guide the robot to the global goal, which is unknown to the robot, while the robot guides the human around nearby obstacles via a local planner. Experimental results in simulation and on hardware show that our controller is robust to external forces, and our seeing-eye system can accurately detect force direction. We demonstrate our full seeing-eye robot system on a real quadruped robot with a blindfolded human. The video can be seen at our project page: https://bu-air-lab.github.io/guide_dog/

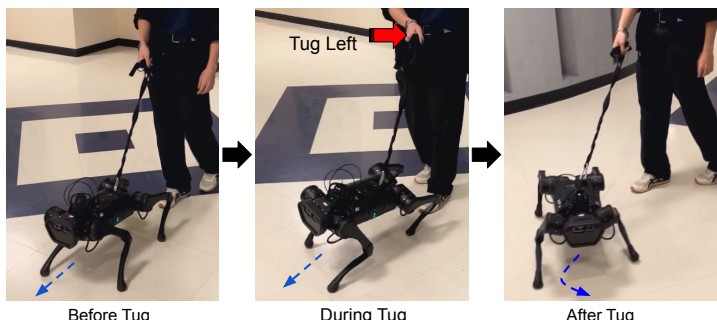

Figure 1: Tugs on the seeing-eye robot are used as navigation cues during blindfolded navigation.

## 1 Introduction

In the typical seeing-eye dog (also known as guide dog) setting, a human holds a rigid handle attached to a harness on a dog. The dog will safely guide the human around nearby obstacles, while the human can tug on the dog to indicate which general direction to move in. Thus, the human has some idea of where they are, and is capable of making high-level navigation decisions, while the dog can better sense its immediate surroundings for obstacle avoidance and locomotion. Guide dogs have been shown to improve the lives of visually impaired people, via increasing independence, confidence, companionship, and mobility [1]. Unfortunately, guide dogs need roughly two years of training, and cost over $50,000 USD per dog [2]. Despite efforts to reduce the production cost of guide dogs [3], their supply is still significantly lower than demand.

7th Conference on Robot Learning (CoRL 2023), Atlanta, USA.

To better meet this demand, and potentially improve performance, seeing-eye robot systems are being developed [4, 5, 6, 7, 8, 9]. There is a growing interest in the development of systems of this type, which have undergone various human studies in order to measure the extent to which they are compatible with visually impaired humans, and their level of societal acceptance [10, 11, 12]. While these seeing-eye robot systems have successfully demonstrated navigation tasks, none of them have considered settings involving human tugs, which is very common for seeing-eye dogs. It's important for seeing-eye robots to be robust to human forces, as the human is constantly holding a rigid handle directly attached to the robot, and large forces can cause the robot to stray from the optimal path, or even fall over. Furthermore, a typical method of communication between a human and a real guide dog is through tugs. This makes force estimation useful for a seeing-eye robot, as knowledge of the direction and magnitude of the applied force can be used to better facilitate navigation according to the human's intentions (see Figure 1).[1]

To address the above mentioned issues in human-robot communication, we develop a novel seeing-eye robot system which is robust to external forces, and estimates the magnitude and direction of these forces to determine which navigation actions to take for human-robot co-navigation. We achieve this by simultaneously training a locomotion controller via RL, and a force estimator via supervised learning. Our locomotion policy is trained over simulated tugs by sampling different base velocities which are suddenly applied to the robot [13]. These tugs serve as labels for training our force estimator, and are estimated during deployment.

While the controller is running on a real robot, the force estimator estimates the direction and magnitude of force the human applies. These force estimates are computed exclusively from sensors onboard the robot (joint encoders and IMU). From force estimates, the robot detects when and in what direction the human tugs occur. This informs the robot which direction to go at a global level, while a local planner using information from a LIDAR sensor is used to navigate the immediate environment. Different from existing seeing-eye robot systems that require major hardware upgrades, e.g., customized traction device [6], or button interface [7], our seeing-eye system needs is compatible with any attachable leash or handle.

Our main contributions include the following:

1. The first seeing-eye robot system which takes directional cues via human tugs, while also safely navigating the immediate environment.

2. A force tolerant locomotion controller, jointly trained with a force estimator which can estimate the magnitude and direction of human forces.

3. Experimental results in simulation and on hardware to evaluate the robustness of our locomotion controller and accuracy of our force estimator.

4. Demonstration of our seeing-eye robot system in an indoor environment with a blindfolded human.

## 2 Related Work

Various seeing-eye robot systems have been demonstrated for the task of blindfolded navigation. Among these works, one of them considers an MPC-based motion planner for a wheeled robot [4]. Another considers an optimization based approach for path planning, which models a taut or slack leash [5]. A third system designs an adjustable leash, and optimizes for human comfort during navigation [6]. These works all stray from real-world guide dog settings, in part because they assume the robot has full knowledge of the destination beforehand, and that the human does not need to communicate with the robot during navigation. In real-world settings, the human must decide on which high-level navigation actions to take, and communicate these actions to the robot.

---

[1]The term of "Seeing-eye robots" has been used by researchers that refer to quadruped robots to guide visually-challenged people [26]. Although our robot doesn't use vision, it serves as a seeing-eye platform through its Lidar sensors for navigation and obstacle avoidance.

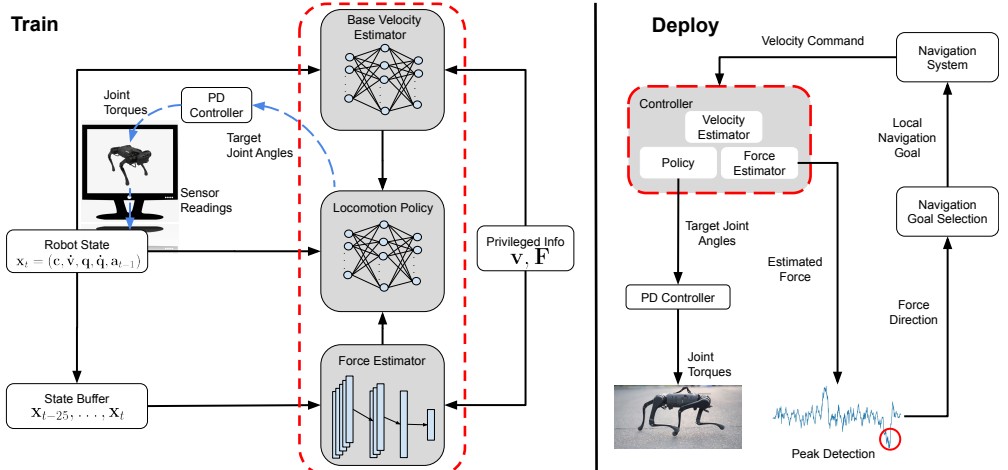

Figure 2: Overview of our approach. Our locomotion controller (circled in red) contains a velocity estimator, force estimator, and locomotion policy, all of which are trained in simulation. The base velocity estimator and force estimator are trained via supervised learning, using privileged information from the simulator as labels. The locomotion policy is trained via RL, and outputs target joint angles to a PD controller which converts them to joint torques which are directly applied to the robot. During deployment, our locomotion controller estimates external force at each time step. Force direction is derived from peaks in the estimated force signal. The direction of force determines the next local navigation goal for our navigation system to take, which returns velocity commands to our controller.

More similar to our work, other approaches consider settings where the human communicates high-level directions to the robot during navigation [7, 8, 9]. However, the medium of communication in these works are either a custom designed handle with buttons [7], predefined directional actions prior to navigation [8], or verbal commands [9]. In our work, the human communicates directional cues via tugging, and is compatible with any rigid connection to the robot. Additionally, none of these works consider human forces being applied to the robot, making their systems unresponsive or susceptible to failure upon navigation under human forces.

Numerous quadruped locomotion controllers have been developed, many of which are robust to external forces [14, 15, 16, 17, 18, 19, 20, 21, 13, 22, 23, 24]. These works do not explicitly estimate the applied forces, as the focus of these works is to develop controllers that are generally robust to varying environments. In our work, we explicitly estimate forces to the robot generated by human tugs, in order to communicate the human's navigation intentions with the robot.

## 3 Method

In this section, we present our seeing-eye robot system that is robust and responsive to external forces from human tugs. Figure 2 presents an overview of how we train our locomotion controller, and deploy for seeing-eye navigation.

### 3.1 Locomotion Controller

We train our locomotion controller via RL, which models environments as a Markov Decision Process (MDP). An MDP is defined as $M = (S, A, T, R, \gamma)$, where $S$ is the set of states, $A$ is the set of actions, $T : S \times A \times S \to [0, 1]$ is the transition function which outputs the probability of reaching state $s'$ given state $s$ and action $a$, $R : S \times A \times S \to \mathbb{R}$ is the reward function which returns feedback from taking action $a$ from state $s$ and ending up in state $s'$, and $\gamma \in [0, 1]$ is the discount factor which determines how valuable future reward should be considered in comparison to immediate reward.

We define a robot state at time $t$ as $\mathbf{x}_t = (\mathbf{c}, \dot{\mathbf{v}}, \mathbf{q}, \dot{\mathbf{q}}, \mathbf{a}_{t-1})$, where $\mathbf{c} = (c_x, c_y, c_\omega)$ is the commanded base linear and angular velocity, $\dot{\mathbf{v}}$ is the base acceleration, $\mathbf{q}$ and $\dot{\mathbf{q}}$ are the joint angles and velocities respectively, and $\mathbf{a}_{t-1}$ is the action taken at time $t - 1$. Actions are target joint angles, which are

Table 1: All terms of the reward function our locomotion policy is trained on. $\mathbf{v}$ refers to base velocity, $\mathbf{c}$ refers to commanded linear and angular base velocity, $\omega$ refers to base angular velocity, $\tau$ refers to joint torques, $\dot{\mathbf{q}}$ refers to joint velocities, $\mathbf{t_{air}}$ refers to each foots air time, $\mathbf{a}$ refers to an action, and $dt$ refers to the simulation time step.

| Term Description | Definition | Scale |
|---|---|---|
| Linear Velocity $x, y$ | $exp(-\|\mathbf{c}_{x,y} - \mathbf{v}_{x,y}\|^2/0.25)$ | $1.0dt$ |
| Linear Velocity $z$ | $\mathbf{v}_z^2$ | $-2.0dt$ |
| Angular Velocity $x, y$ | $\|\omega_{\mathbf{x,y}}\|^2$ | $-0.05dt$ |
| Angular Velocity $z$ | $exp(-(\mathbf{c}_\omega - \omega_z)^2/0.25)$ | $0.5dt$ |
| Joint Torques | $\|\tau\|^2$ | $-0.0002dt$ |
| Joint Accelerations | $\|(\dot{\mathbf{q}}_{last} - \dot{\mathbf{q}})/dt\|^2$ | $-2.5e-7dt$ |
| Feet Air Time | $\sum_{f=1}^{4}(\mathbf{t_{air,f}} - 0.5)$ | $1.0dt$ |
| Action Rate | $\|\mathbf{a}_{last} - \mathbf{a}\|^2$ | $-0.01dt$ |

converted to torques via a PD controller. The reward function encourages tracking commands $\mathbf{c}$ while minimizing energy consumption and large action changes [13] – fully defined in Table 1.

Our locomotion policy also takes the base velocity and external force vector as input, to more easily learn to track commands $\mathbf{c}$ and respond to external forces. These variables are not easily estimated through robot sensors. Thus, we train state estimators via supervised learning over privileged information, which has been shown to be more effective than classical methods, e.g., Kalman Filters [25]. It is a common setting to learn with privileged information in simulation, and insert a corresponding state estimator in real-world deployment. In line with those systems, we train a velocity estimator and force estimator for the real robot to bridge the sim-to-real gap in locomotion policy learning. These estimators are trained jointly with the locomotion policy, where $\mathbf{v} = (v_x, v_y, v_z)$ is ground truth base velocity, and $\mathbf{F} = (F_x, F_y, F_z)$ is ground truth external force, both obtained as privileged information from the simulator.

Two variations of forces are applied to the robot during training. One variation is small and frequent backward pushes, which are designed to simulate a human following the robot with a taut leash, where a human incidentally applies frequent small backward forces on the robot as they are being guided. The other variation is larger, less frequent pushes occurring in any direction. These pushes are designed to simulate human directional tugs, in which the human intentionally tugs the robot to communicate the direction they want to move in. The force estimator is only trained on data from the second variation of tugs, as we do not want to detect the small, incidental backward tugs which naturally occur during guided navigation.

The base velocity estimator is a multilayer perceptron (MLP), whose parameters are updated over the same data as the locomotion policy. To estimate forces, we find it helpful to access a history of states, to better capture the robot's behavior over the duration of the applied force. Thus, similar to training adaptation modules [17], our force estimator uses 1-D convolutional layers to capture temporal relationships between states. Force estimator parameters are updated less frequently than the base velocity estimator and locomotion policy, because most time steps do not include external forces. Thus, most of the labels for our force estimator includes zero vectors, indicating that no force was applied at those time steps. This causes imbalanced training data, which we resolve by only training the force estimator when the training data contains nonzero forces. We then further re-balance this to ensure at most 20% of the force estimator's training samples include zero vectors as labels.

## 3.2 Seeing-eye Robot Navigation

To perform navigation tasks with our seeing-eye robot, we need to estimate when and in what general direction a force is being applied. This is done by running peak detection [26] on the previous 200 time steps of the estimated force signal $F_y$, at a rate of 2Hz. In this work, we only analyze $F_y$, to determine whether a left or right tug has occurred. A peak in the signal indicates that the force estimator detected a significant external force applied to the robot. Thus, if a peak is detected within

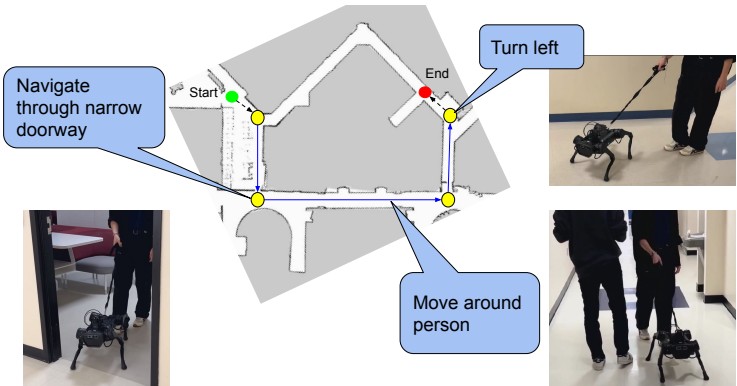

Figure 3: Map of our navigation environment. Yellow circles correspond to *decision points*, where the human needs to decide which direction to move in via tugging. The blue lines indicate the path the human took in our demonstration.

the past 50 time steps, then we consider it as a recent tug applied by the human. Positive peaks correspond to left tugs, while negative peaks correspond to right tugs.

Local navigation goals are then selected based on the robot's location, orientation, and tug direction. A domain expert manually labels a map with *decision points*, where the human must decide which direction to go in next, based on the direction they came from, and their tug direction. A labelled map of the hallway domain we demonstrate our system on can be seen in Figure 3.

The local navigation goal is then sent to our navigation system, which uses a LIDAR sensor to localize itself on the map via AMCL [27], and compute local plans via DWA [28]. The local planner returns velocity commands **c** to our controller, which our controller tracks.

## 4    Implementation Details

We use Isaac Gym [29] physics simulator, and train our controller with 2048 robots in parallel [13]. Our locomotion policy is trained via PPO [30], while our base velocity estimator and force estimator are trained via supervised learning, with Mean Squared Error loss. Our locomotion policy and base velocity estimator are both MLPs with three and two hidden layers respectively, while our force estimator is a convolutional neural network with three 1-D convolutional layers, which makes predictions over the past 25 time steps. The PD controller converts target joint angles to torques with proportional gain set to 20 and derivative gain set to 0.5. The policy is queried at 50Hz, and control signals are sent at 200Hz. New velocity commands are sampled after each episode, where $c_x$, $c_y$, and $c_\omega$ are sampled uniformly from [-1,1]. An episode terminates when any link other than a foot touches the ground, the base height is below 0.25 meters, or the episode has lasted 20 seconds.

To better facilitate sim-to-real transfer, we train over RANDOM_UNIFORM_TERRAIN which increases in difficulty based on a curriculum [13]. We also include noise in observations, and domain randomization over different surface frictions.

We add random external forces in our environment, to improve robustness of our locomotion policy and collect data to train our force estimator. The small and frequent backward pushes occur every 0.6 seconds, have a duration of 0.1 seconds, and sets the base velocity to 0.25 m/s backward. Meanwhile, the large and infrequent pushes used to train the force estimator occur every 3 seconds, have a duration sampled from [0.24, 0.48] seconds, and sets the base velocity to a vector sampled from $F_x \in [-0.75, 0.75]$, $F_y \in [-0.75, 0.75]$, and $F_z \in [0, 0.1]$. Note that forces from **F** are implemented as spontaneous updates in base velocity.

## 5    Experiments

We design experiments to evaluate the robustness of our controller, and accuracy of our force estimator. Although we are able to learn a force estimator in simulation, we could not directly evaluate

Table 2: Our learned controllers fell significantly less frequently, and better tracked velocity commands under external forces when compared to an MPC controller. Including the output of the force estimator in the state marginally improved robustness. The large variance in drift is caused by the large range of force strengths and directions we sample from.

| Controller | Proportion Fell | Drift from Trajectory |
|---|---|---|
| MPC | 0.5990 | $1.1256 \pm 0.5908$ |
| Learned No Est | 0.1904 | $0.6824 \pm 0.5447$ |
| Learned Est | **0.1762** | **$0.6790 \pm 0.5309$** |

it in the real world due to the missing ground-truth values. Instead, we chose to evaluate tug detection (LEFT, RIGHT, and NONE) on the real robot, where the participants followed our instructions, and hence ground truth was available. We then demonstrate our full seeing-eye system via guiding a blindfolded human.

## 5.1 Force Tolerance Evaluation

To evaluate whether our learned force controller is actually robust to external forces, we run experiments in simulation where we apply random forces to the base of the robot. In each trial, a single force in a random direction is applied, whose duration is sampled from [0.25, 0.5] seconds, and strength is sampled from [25, 100] Newtons. Meanwhile, the robot is commanded to walk forward at 0.5 meters/second, for a duration of five seconds.

We run this experiment on three different types of locomotion controllers, for 1000 trials each. The controllers include a commonly used MPC controller [31], a variant of our learned controller which does not consider estimated force in its state (referred to as *Learned No Est*), and our controller described in Section 3.1 (referred to as *Learned Est*). All controllers are deployed in PyBullet [32].

We measure how frequently the robot fell across all trials (Proportion Fell), and how far the external force caused the robot to drift from its current trajectory on average (Drift from Trajectory). We consider a robot to have fell if a non-foot part of the robot touches the ground. Results are reported in Table 2, which indicate that our learned controllers fell significantly less frequently, and better maintained velocity tracking under external forces than the MPC controller. Including estimated forces in the state appears to marginally improve robustness. Note that for our two types of learning-based controllers, we train over five different random seeds each and average the results.

## 5.2 Force Estimation Evaluation

We evaluate the accuracy of our force estimator through experiments in simulation, and on hardware.

### 5.2.1 Simulation

We deploy our learned controllers in PyBullet with a velocity command of 0.5 meters/second, and a single external push per trial, for 1000 trials. Each push has a force whose x-component is sampled from [-50, 50] Newtons, and a y-component which is at a fixed magnitude, and random direction (either left or right). There are three possible classes the force estimator can predict over: {LEFT, RIGHT, NONE}. Each trial includes 150 time steps and takes 3 seconds. In each trial, the force estimator is queried every 25 time steps, or six total queries per trial. Thus, in each trial, the force estimator makes a total of six predictions. A trial is deemed correct if one of the six queries matches the force direction being applied in the simulator.

In order to evaluate whether force direction can be estimated through only a history of base velocities, we train a baseline force estimator which only makes force esti-

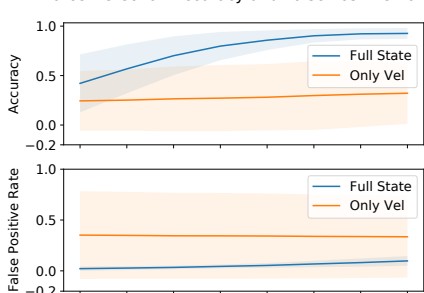

Figure 4: We report the accuracy and false positive rate of our force estimators, given forces of varied strength. The shaded region indicates the standard deviation between the five policies trained over five different random seeds.

mates based on a history of ground-truth base velocity and base velocity commands. We refer to this baseline as *only vel*, while our force estimator trained over the full state and described in Section 3.1 is referred to as *full state*.

In Figure 4, we report the accuracy and false positive rates of our force estimators over varying force strengths. Accuracy is computed by dividing the number of trials in which the force estimator predicted the correct force direction at any of the six queries in the trial, by the number of total trials. Computing accuracy alone in this manner is not informative enough, because it is possible to achieve high accuracy by predicting LEFT at one point in the trial, and RIGHT at a different point within the same trial.

Thus, we also consider the false positive rate, which we compute by dividing the number of extra forces (LEFT or RIGHT predicted when ground truth is NONE) predicted during the trial, by the number of times the force estimator is queried (every 25 time steps). A high false positive rate corresponds to the force estimator oftentimes predicting forces when they do not occur. As force strength increases, our estimators achieve a higher accuracy while maintaining a relatively low false positive rate. Results indicate that knowledge of the full state is significantly beneficial in estimating force direction, when compared to a force estimator which is only trained over base velocity information.

### 5.2.2 Hardware

When a human tugs our seeing-eye robot with sufficient force, the base of the robot will momentarily accelerate in the direction of the tug. Thus, one might wonder why we do not consider accelerometer signals to detect tug direction, rather than train a force estimator. In this sub-section, we validate the usefulness of our estimated force signals, which we compare to accelerometer readings.

In this experiment, we command a real Unitree A1 robot to move forward at 0.5m/s, while a human participant tugs left after a few seconds of forward locomotion, followed by a right tug after another few seconds of locomotion. This trial is performed by four human participants, three of which have no prior experience in operating this tugging interface. Each participant is a robotics researcher in a university lab. The three participants with no prior experience with this system were given a demonstration of an example trial, before completing their own trials. In total, 42 trials were conducted, and data was collected from 40 trials (two trials were removed due to the robot falling over and data not being saved). Of these 40 trials, each participant performed ten of them.

Forces are detected every 25 timesteps, and can be predicted as one of three classes. We compute the accuracy and false positive rate for force detectors using the estimated force signal, compared to force detection using the signal directly from the accelerometer on the robot. Accuracy is computed as the percentage of trials which contain a LEFT force prediction before the halfway point of the trial, and a RIGHT force prediction after the halfway point of the trial. False positive rate is the average percentage of false positive forces being predicted across all trials. A force prediction is considered as a false positive if it is either LEFT or RIGHT, and does not correspond to the first expected LEFT force or later expected RIGHT force.

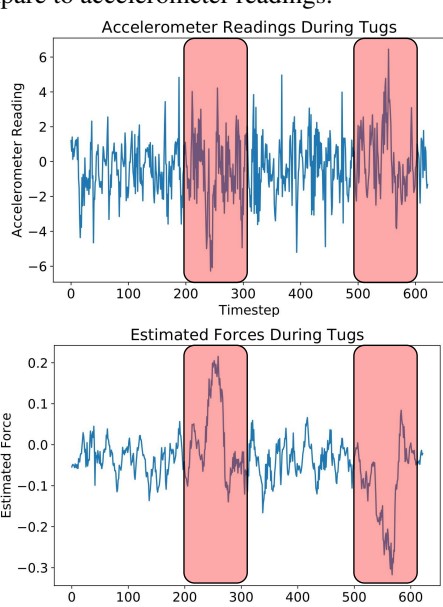

Figure 5: Measured acceleration (top) and estimated force (bottom) during a single trial. Tugs are denoted by red boxes.

Results are reported in Table 3, where we find the force estimator more accurately predicted the correct forces, with fewer false positives compared to predictions from raw accelerometer signals. Both methods of detecting tug direction (accelerometer and force estimator) perform worse for be-

Table 3: Force estimation via accelerometer readings vs force estimator signal.

| Participant Type | Method | Accuracy | False Positive Rate |
|---|---|---|---|
| Expert (10 trials) | Accelerometer | 0.20 | 0.2149 |
| | Force Estimator | 1.00 | 0.0442 |
| Beginner (30 trials) | Accelerometer | 0.13 | 0.1906 |
| | Force Estimator | 0.70 | 0.0498 |

ginner participants, indicating that adding diverse tugging styles is important for a more complete evaluation of force estimators.

In Figure 5, we plot the raw signals from the accelerometer and force estimator from a single trial. We find the accelerometer signal is more noisy than the estimated force signal, which we hypothesize is because the force estimator has access to other sensor information along with accelerometer readings. Note that our force estimator is co-learned with the locomotion policy. The locomotion policy takes the estimated force as input during training, and the states generated from the policy are used to train the force estimator. We believe this co-learning mechanism leverages the additional sensor information to help the force estimator outperform the tug detection from raw accelerometer readings.

## 6    Hardware Demonstration

We demonstrate our full seeing-eye robot system on a Unitree A1 robot, in an indoor hallway environment (see link in Abstract). Our locomotion policy and force estimator are deployed on hardware without any additional fine-tuning.

In this demonstration, a human is blindfolded, and holding a taut leash attached to the robot. The human desires to reach some particular goal location, and chooses the rout to take by tugging the robot at decision points, while the robot autonomously avoids obstacles (including boxes, narrow doorways, and another human) along the way. Decision points occur at intersections in the hallway, where the human needs to decide in which direction they want to go. Similar to the setting in [7], the blindfolded human is not familiar with localizing himself without vision, so another sighted human verbally indicates when a decision point is approaching.

This demonstration indicates that our locomotion policy and force detector are transferable to hardware, and can cooperate with a local planner to enable blindfolded navigation. Thus, successful human-robot communication occurred, such that the human avoided all obstacles while the robot navigated to the desired goal location through detecting human tugs at decision points.

## 7    Discussion

**Limitations and Future Work**    While our seeing-eye system can avoid obstacles and select routes at a course level in indoor hallway environments, real guide dog settings include outdoor environments, and situations with many possible directions to navigate in. In future work, researchers can leverage methods to determine which paths are traversable [33], and train force detectors which can estimate the direction of force at a finer level. Another future direction is to incorporate intelligent disobedience, which refers to rejecting a human's navigation decision if unsafe [34]. Additionally, The evaluation can be further improved by replacing sighted people with those with visual impairments in the experiments. Finally, our robot's locomotion speed is relatively low. It might be the human tugs, the learned locomotion policy, or both causing the low speed. Further investigation into those factors can potentially lead to very interesting future research and higher-speed seeing-eye robot systems.

**Conclusion**    We train a locomotion controller which is tolerant to human tugs, and can estimate the direction of external forces. We evaluate the robustness of our controller, and accuracy of our force estimator through experiments in simulation and on hardware. Finally, we demonstrate our controller on a real robot for the task of blindfolded navigation, where a blindfolded human is successfully guided to a destination, while giving directional cues through tugging on the robot.

## Acknowledgement

This work has taken place at the Autonomous Intelligent Robotics (AIR) Group, SUNY Binghamton. AIR research is supported in part by grants from the National Science Foundation (NRI-1925044), Ford Motor Company, OPPO, and SUNY Research Foundation.

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
