# OpenReview forum: "Seeing-Eye Quadruped Navigation with Force Responsive Locomotion Control"
_robot-learning.org/CoRL/2023/Conference — CoRL 2023 Poster_

### Official Review · Reviewer_5maN · 2023-06-21

**Confidence:** 5
**Originality:** Fair
**Technical Quality:** Fair
**Clarity Of Presentation:** Excellent
**Impact:** 2

**Recommendation:**

Weak Reject: I recommend rejecting the paper, but will not argue for my recommendation if the majority of other reviewers have a different opinion.

**Review:**

**Strengths**
- The work provides a reasonable interface for commanding legged locomotion. The hardware experiment on the specified setup is convincing.
- It argues for the value of the external force estimation by explicitly comparing it with the results from accelerometer readings.
- The distinguishment of two kinds of external forces applied to the robot makes a good point.

**Weaknesses**
- The work has limited novelty on the technical side. The RL locomotion framework follows an existing open-sourced work and the addition of the training of the estimators is also not new. The work seems to combine two existing works (Learning to Walk in Minutes + RMA) to me with a well-wrapped estimated force command interface.
- While the work seems to be motivated for significant help in guiding visually impaired people, the final results do not seem convincing. As the authors mention in the limitation, the setting assumes prescribed decision points in a known map, considers only binary tug directions, works under specific running velocity settings, and has limited traversability. Summing up these points, the work seems to have too many simplifications to stand for its argument to help visually impaired people. More results on an extended capability are expected, especially given the limited algorithmic contribution.
- The setup does not seem self-contained. At decision points (which are already known following the assumption in the work), why not directly set such binary commands rather than going through a tug-force interface? And if there is a sighted person dictating the commands, why would we even need the robot?

**Quality Of The Limitations Section:**

Limitations are addressed clearly

**Questions For Rebuttal:**

- The paper did not touch on how the navigation system selects the goal and determines the commanded velocity after receiving detected tug forces. More information on the navigation module is needed.
- The paper did not argue clearly the reason for using the base velocity estimator. More information is needed.
- The results in Table 2 are not convincing given the large variance in the drift column. Please either explain the reason or consider other evaluation metrics.
- More technical novelty or better deployment capability is expected. Please illustrate the challenges you solved, which would have not been solved by using other methods.
- It would be more convincing to show ablation on external force sampling strategy during training, especially for the need of considering small back-pushing forces.
- Please address the weaknesses above accordingly.

**Robotics Focus:**

Sufficient demonstration on hardware

**Summary Of Paper:**

The authors proposed a locomotion algorithm for seeing-eye robots where external tug forces are used to determine high-level local navigation directions, which further specify low-level velocity targets. The novel proposal of an online external force estimation module provides a reasonable replacement for accelerometer reading which can be noisy and unreliable in practice. The hardware experiments showed convincing results and effectiveness in guiding visually impaired people by using the proposed system.

**Summary Of Recommendation:**

I acknowledge the authors' insights into the potential of the work in helping visually impaired people. And I acknowledge the fact that the proposed system works well in limited scenarios under specific assumptions. This work can be viewed as the first step in achieving the eventual goal it argues for. However, as a research paper, I would expect it to have either technical novelty or strong application performance. As the paper is developed by combining existing works, I did not see strong evidence of novel technical contribution. In this case, to be considered to be accepted, the paper should show strong and convincing application system performance with high generality and robustness. Nevertheless, the proposed system is shown effective only in very limited domains under strong simplifications and assumptions. I would suggest the authors bring more convincing results of extended capabilities at the rebuttal phase.

---

### Official Review · Reviewer_wGQP · 2023-07-03

**Confidence:** 5
**Originality:** Very Good
**Technical Quality:** Very Good
**Clarity Of Presentation:** Excellent
**Impact:** 4

**Recommendation:**

Strong Accept: I recommend accepting the paper and will argue for my recommendation even if other reviewers hold a different opinion.

**Review:**

This is really a nice and inspiring paper -- mainly for the application and the promotion of diversity in the area of legged robots, than for the methodology itself. Nevertheless, a supervised force estimation and implementation on the real robot makes the paper really nice. The locomotion is basic with RL (and maybe not the focus of the paper). The focus is the estimation of the force to affect the local plan decision when navigating globally.

**Quality Of The Limitations Section:**

Limitations are addressed clearly

**Questions For Rebuttal:**

There are two questions for the authors:
1. why locomotion was learned instead of using Unitree's (or other robots) MPC locomotion and learn the force estimation only?
2. how did you evaluate the correct force estimation on the real hardware without ground truth?

**Robotics Focus:**

Sufficient demonstration on hardware

**Summary Of Paper:**

This is the first paper to consider human tugs for guiding a guide robot dog.

**Summary Of Recommendation:**

Overall a nice application of machine learning in robotics, for an inspiring paper and purpose.

---

### Official Review · Reviewer_Pxez · 2023-07-14

**Confidence:** 4
**Originality:** Good
**Technical Quality:** Good
**Clarity Of Presentation:** Very Good
**Impact:** 3

**Recommendation:**

Weak Accept: I recommend accepting the paper, but will not argue for my recommendation if the majority of other reviewers have a different opinion.

**Review:**

The system is straightforward and well executed, with sufficient evaluations in both simulation and the real robot.

A main novelty of the work is the use of a force estimator, which can both help improve the robustness of the policy as well as help the user to provide directional cues.

However, the arguments against directly using an accelerometer signal are a little bit circular. In the paper, the authors state: ". For example, the policy might learn to make certain joint movements when a tug occurs. Our force estimator can learn to associate these joint movements with the corresponding tug direction, while the accelerometer has no such access to this information. "

Here are some questions about this argument:

1. The policy makes certain joint movements when a tug occurs. But what triggers those movements? Because of the tug, not the other way around, i.e., the policy first detects the tug. Then how does the policy detect it? It is kind of like a chicken-egg argument, hope this makes sense.

2. My guess will be the policy detects the tug by observing changes in velocity signal that deviates from what will happen if there are no tugs. Then it is entirely possible to detect tugs from body velocity only.

Since the force estimator is the main contribution here, I think it is important to investigate this issue further.

**Quality Of The Limitations Section:**

Limitations are addressed clearly

**Questions For Rebuttal:**

Given my concern above, it will be nice to have a baseline where the force estimator is only trained with base information instead of the full state.

It will also be interesting to filter the raw accelerator readings to see if the signal becomes clearer.

Another comparison will be to compare the learned base velocity estimator with a standard velocity estimator based on Kalman Filter. A Kalman Filter-based estimator can also be used to generate acceleration signals to detect tugs for comparison.

**Robotics Focus:**

Sufficient demonstration on hardware

**Summary Of Paper:**

This paper presents a system that allows a quadruped robot to guide visually impaired humans. Directional cues can be taken from human tug via the use of the force estimator. Demonstration of the system is done in both simulation and the real robot.

**Summary Of Recommendation:**

This paper builds a nice system to use quadrupedal robots as guide dogs. However, because of my concern about the importance of a learned force estimator, I recommend weak rejection for now. If additional experiments can show its necessity, I am willing to raise my score.

---

### Official Review · Reviewer_BKt3 · 2023-07-16

**Confidence:** 5
**Originality:** Good
**Technical Quality:** Very Good
**Clarity Of Presentation:** Excellent
**Impact:** 3

**Recommendation:**

Strong Accept: I recommend accepting the paper and will argue for my recommendation even if other reviewers hold a different opinion.

**Review:**

The paper is well organized and the writing is clear. The presented methods are sound. The paper is interesting and original because the authors use force estimation from human tugs to give direction commands to a seeing-eye robot where such human-robot interaction had drawn little attention in the literature. Using simulation and real-world experiments the authors illustrate that the robot can learn to facilitate and assist navigation according to the human’s intentions. However, the robot locomotion speeds remain slow and might affect the applicability of the proposed approach in real settings.



**Quality Of The Limitations Section:**

Limitations are addressed clearly

**Questions For Rebuttal:**

It would be interesting to see the robot navigate at higher speeds.

**Robotics Focus:**

Sufficient demonstration on hardware

**Summary Of Paper:**

This work presents a seeing-eye quadrupedal robot capable of taking directional cues from human tugs via force estimation, simultaneously offering safe traversal in an indoor environment. The authors are motivated by the lack of human tug input to the robot for directional purposes.  The authors present a scheme allowing the robot to learn to estimate the magnitude and direction of the forces due to human tugs. The paper presents simulation results and real-world tests to show their proposed approach.

**Summary Of Recommendation:**

Motivated by the limitations on the use of human tugs to give direction cues to seeing-eye robots, the authors utilized learning techniques on force estimation and simultaneous locomotion generation from intermittent tugs. The presented approach to guidance robots is novel and future robots might benefit from the contribution.

---

### Official Review · Reviewer_SGaq · 2023-07-25

**Confidence:** 4
**Originality:** Good
**Technical Quality:** Good
**Clarity Of Presentation:** Very Good
**Impact:** 3

**Recommendation:**

Weak Accept: I recommend accepting the paper, but will not argue for my recommendation if the majority of other reviewers have a different opinion.

**Review:**

This paper has extend the quadruped locomotion controller with force estimator and noveally use the estimator not only in the locomotion policy but also as a form of user input.

This work is well documented with details.
* The model arch and reward is clearly illustrated and documented. Training methods is also well documented.
* The idea of using force estimation not only for a more robust but also slightly improves tolerance of force.

A few places that can be improved.
* The force estimation part is not quantitatively evaluated on real robot.
* Some metrics of the experiments are not clearly described. For example, in table 2, the "proportion fell" for "learning no est" is 0.1904 in 1000 trials. It is not very clear how the metrics is measured.
* The force tolerant part of the paper is not compared with other work in "related work".
* While the paper is titled "Seeing-eye" and is for visually impaired people, the paper's content requires human to "guide" the robot. This can be misleading.

**Quality Of The Limitations Section:**

Limitations are addressed clearly

**Questions For Rebuttal:**

1) The force estimation work can be better supported by a quantitatively evaluated on real robot. This will address the sim to real question.

2) Please clarify how the metrics are computed. Otherwise some of the numbers are difficult to interpret.

3) No vision is involved in this work although the paper is titled "Seeing-eye", this is a bit misleading. Consider changing the title or clarify at the beginning of the paper.

**Robotics Focus:**

Sufficient demonstration on hardware

**Summary Of Paper:**

This paper documented co-training of a legged robot locomotion policy and external force estimator and using the force estimator as a form of user input in navigation context.

In the paper the method of training is documented in detail (reward design, model arch, task setup). The results include force tolerance in sim and force estimation evaluation in both simulation and real.

**Summary Of Recommendation:**

Recommend accept the paper if the rebuttal questions are addressed.

---

### Author Response · Authors · 2023-08-13
**General Response**

We thank the reviewers for the valuable feedback. All reviewers agree that our work was presented clearly (including three “Excellent” and two “Very Good” scores under “Clarity Of Presentation”). Before responding to each reviewer individually, we would like to highlight the most notable new improvements that we have made to our paper during the rebuttal phase:

1. A new baseline (called “Only Vel”) has been added into the evaluation of our learned force estimator in simulation, as shown in Figure 4 in Section 5.2.1. “Only Vel” trains only on a history of base velocities and corresponding base velocity commands, rather than a history of the full state. This is to address the comments from Reviewer Pxez.
2. Three new human participants have been recruited for evaluating our force estimator in the real world. The new results are described in Table 3 in Section 5.2.2. The previous version included only ten trials from one participant, while the updated version includes 42 total trials. Note that all three new participants have never used this seeing-eye system prior to the trials, which has added a greater diversity in tug styles to our evaluation as well as a performance reduction compared with the previous results. This is to address the comments about evaluating the force estimator from Reviewer SGaq.

We have updated the paper, which is attached to each individual response. Changes to the text are highlighted in blue font.

---

### Decision · Program_Chairs · 2023-08-30

**Decision:**

Accept (Poster)

**Comment:**

The AC is excited to recommend accepting this work on the given positive reviews. The paper presents a novel force-responsive locomotion control system, which is well-validated. Please resolve the reviewers' comments as much as possible at the final stage. It will also be great to have a user study with the actual user population. Congratulations!